



**Title: Lichen species across Alaska produce highly active and stable ice nucleators**

**Authors:** Rosemary J. Eufemio[1,2], Ingrid de Almeida Ribeiro[3], Todd L. Sformo[4], Gary A. Laursen[5], Valeria Molinero[3], Janine Fröhlich-Nowoisky[6], Mischa Bonn[7], and Konrad Meister[1,2,7*]

**Affiliations:**

[1]Department of Chemistry and Biochemistry, Boise State University, Boise, ID 83725, USA

[2]Biomolecular Sciences Graduate Programs, Boise State University, Boise, ID 83725, USA

[3]University of Utah, 84112 Salt Lake City, UT, United States

[4]Institute of Arctic Biology, University of Alaska Fairbanks, Fairbanks, AK 99775, USA

[5]High Latitude Mycological Research Institute, University of Montana, Missoula, MT 59801, USA

[6]Max Planck Institute for Chemistry, 55128 Mainz, Germany

[7]Max Planck Institute for Polymer Research, 55128 Mainz, Germany

*Correspondence to:* Konrad Meister (konradmeister@boisestate.edu)

**Abstract**

Forty years ago, lichens were identified as extraordinary biological ice nucleators (INs) that enable ice formation at temperatures close to 0°C. By employing INs, lichens thrive in freezing environments that surpass the physiological limits of other vegetation, thus making them the majority of vegetative biomass in northern ecosystems. Aerosolized lichen INs might further impact cloud glaciation and have the potential to alter atmospheric processes in a warming Arctic. Despite the ecological importance and formidable ice nucleation activities, the abundance, diversity, sources, and role of ice nucleation in lichens remain poorly understood. Here, we investigate the ice nucleation capabilities of lichens collected from various ecosystems across Alaska. We find ice-nucleating activity in lichen to be widespread, particularly in the coastal rainforest of Southeast Alaska. Across 29 investigated lichen, all species show ice nucleation





temperatures above -15°C and ~30% initiate freezing at temperatures above -6°C. Concentration series of lichen ice nucleation assays in combination with statistical analysis reveal that the lichens contain two subpopulations of INs, similar to previous observations in bacteria. However, unlike the bacterial INs, the lichen INs appear as independent subpopulations resistant to freeze-thaw cycles and against temperature treatment. The ubiquity and high stability of the lichen INs suggest that they can impact local atmospheric processes and that ice nucleation activity is an essential trait for their survival in cold environments.

## 1    Introduction

The formation of ice is thermodynamically favored at temperatures below 0 °C, but the crystallization is kinetically hindered. As a result, pure water droplets can be supercooled to temperatures as low as −38°C, below which homogeneous ice nucleation occurs (Koop et al., 2000). In natural systems, water usually freezes in a heterogeneous process facilitated by the presence of particles that serve as ice nucleators (INs). INs can be of biotic and abiotic origins (Maki and Willoughby, 1978; Wilson et al., 2003; Murray et al., 2012) and play fundamental roles in high-altitude cloud formation, in triggering precipitation in mixed-phase clouds, and assist in the survival of freeze-tolerant organisms (Zachariassen and Kristiansen, 2000). Potent biological INs have been discovered in various life forms, including bacteria, fungi, algae, plants, animals, and lichen (Maki et al., 1974; Kieft and Lindow, 1988; Pouleur et al., 1992; Lundheim, 2002; Fröhlich-Nowoisky et al., 2015). The most efficient and best-characterized biological INs are the plant-associated bacteria *Pseudomonas syringae* (Kozloff et al., 1983; Govindarajan and Lindow, 1988). The ability of *P. syringae* to facilitate ice formation is attributed to specialized ice-nucleating proteins (INPs) anchored to their outer bacterial cell membrane (Govindarajan and Lindow, 1988). *P. syringae* INPs form functional aggregates that are categorized into classes A-C, depending on their activation temperature and assembly size (Turner et al., 1990; Lukas et al., 2022). Class C consists of small clusters of INPs and is responsible for freezing at approximately -7.5°C, while class A is an assembly of class C INs and induces freezing as warm as -2°C. Class B INs are rarely observed and responsible for freezing between ~−5 and ~−7 °C (Schwidetzky et al., 2021b).



Next to bacteria, lichens have been identified as extraordinary INs (Kieft and Lindow, 1988; Honegger, 2007; Moffett et al., 2015). Given the global abundance of lichen (Honegger, 2007), airborne lichen-derived INs have been proposed to contribute to the pool of atmospheric INs and to strongly influence atmospheric processes (Moffett et al., 2015; Creamean et al., 2021). Lichens depend on the atmosphere for moisture and the ability to facilitate ice formation has been proposed to be beneficial to scavenge water vapor in cold environments (Kieft and Lindow, 1988). Of the lichen species previously tested, the majority were ice nucleation active, indicating that IN-activity in lichens is ubiquitous (Kieft and Lindow, 1988; Moffett et al., 2015). Lichen IN activity was primarily attributed to the mycobiont (fungal symbiont), and the photobiont partner was proposed to be relatively inactive (Kieft and Ahmadjian, 1989). Efforts to identify and characterize lichen INs revealed that they are active without lipids and very pH- and heat-stable (Kieft and Ahmadjian, 1989; Kieft and Ruscetti, 1990). Studies on non-lichenized fungi further suggested that proteinaceous compounds are often responsible for their ice nucleation activity (Kieft and Ruscetti, 1990; Lundheim, 2002; Kunert et al., 2019; Fröhlich-Nowoisky et al., 2015; Moffett et al., 2015). However, while non-lichenized fungal species like *Fusarium* have been extensively investigated for their IN activity and overall influence on the environment and atmosphere (Fröhlich-Nowoisky et al., 2015; Kunert et al., 2019; Yang et al., 2022), our understanding of the IN activity of lichenized fungi and the prevalence of ice nucleation among lichen species remains limited (Kieft and Ruscetti, 1990; Henderson-Begg et al., 2009; Moffett et al., 2015).

Here, we surveyed a taxonomically diverse selection of lichen species from across Alaska for IN activity. We used the high-throughput twin-plate ice nucleation assay (TINA) to quantify INs of selected lichen species, and the underlying distribution of INs was determined using numerical modeling of the cumulative freezing spectra. Further, we investigated the stability of lichen INs upon exposure to freeze-thaw cycles and heat treatments, to gain insights into possible atmospheric influences and their usage in cryostorage applications.

## 2 Materials and methods

### 2.1 Sampling





Samples were obtained from four regions in Alaska, USA, between September 2021 and February
90    2022. Nineteen lichen species were collected from rock and tree bark substrates in Juneau, three
in Kodiak, five in Utqiagvik, and two in Fairbanks. These locations encompassed a range of
biomes, from coastal rainforests in Juneau and Kodiak, to Arctic tundra in Utquigvik, and to the
boreal forest in Fairbanks (Simpson et al., 2007). Samples were collected in sterile containers and
stored at -18°C prior to analysis.

95

**2.2 Purification of aqueous extracts**

To prepare extracts for quantitative analysis and temperature experiments, all samples were
washed with pure water (Millipore Milli-Q® Simplicity 185 Water Purification System, Merck
100    KGaA, Darmstadt, Germany) to minimize contamination from external sources of INs. Extracts
were prepared for ice nucleation assays and stability experiments using a standardized procedure
adapted from Kieft (Kieft and Lindow, 1988). First, a 5 mL aliquot of pure water was added to
2 g of lichen. Secondly, the samples were ground to a fine pulp using a mortar and pestle, and
centrifuged at 5000 rpm for 10 min. The supernatants were filtered through 0.22 $\mu$m pore diameter
syringe filters (Millex® Syringe Filter, Merck KGaA, Darmstadt, Germany) to remove cellular
debris. The aqueous extracts were immediately tested for ice nucleation activity using the Vali-
type (Vali, 1971) apparatus and stored at -18°C until systematic analysis. Lichen extracts remained
frozen for up to nine months.

**2.3 Initial screening for ice nucleation activity**

For initial screenings of lichen ice nucleation activity, a Vali-type apparatus was used, which
consists of a temperature-controlled aluminum cold plate (Linkam Scientific Instruments LTD,
United Kingdom). For each sample extract, twenty 1 $\mu$L droplets were cooled at a rate of 3°C
min$^{-1}$ and freezing events were recorded through the optical change in droplet appearance
consistent with freezing. The freezing temperature for each droplet was recorded, and the
temperature at which half of the droplets are frozen, $T_{50}$, is used as a direct measure for the efficacy
of the INs. Snomax (inactivated *P. syringae*, Snomax Int) was used as a positive control, resulting
in a $T_{50}$ of -3.5°C at 1.0 mg/mL, and pure water had at $T_{50}$ of -15°C.




### 2.4 Freezing spectra (number of IN)

The three most active lichen species, *Platismatia herrei*, *Sphaerophorus globosus*, and *Peltigera britannica*, were analyzed using the high-throughput Twin-plate Ice Nucleation Assay (TINA)

(Kunert et al., 2018). In TINA measurements a liquid handling station (epMotion ep5073, Eppendorf, Hamburg, Germany) was used to serially dilute the aqueous extracts in 10-fold increments with pure water, each dilution consisting of 96 (3 $\mu$L) droplets in two 384-well plates. The 384-well plates were cooled at a continuous rate of 1°C min$^{-1}$ from 0°C to -30°C. Freezing temperatures for each droplet were extracted, and the fraction of frozen droplets was calculated

for the different temperatures. The cumulative number of active INs per mass unit of sample ($N_m$) in each dilution was calculated using Vali's equation (Vali, 1971).

### 2.5 Modeling underlying distribution

The Heterogeneous Underlying-Based (HUB) method and associated HUB-backward numerical code (de Almeida Ribeiro et al., 2022) were used to interpret the heterogeneous ice nucleation temperatures obtained from TINA measurements of the lichen samples. The HUB method relies on the assumptions of Vali (Vali, 1971), e.g. that each IN has a distinct nucleation temperature, and that the IN with the highest nucleation temperature is responsible for the freezing of the droplet

in the cooling experiments. The HUB-backwards stochastic optimization procedure was used to determine the underlying distribution of ice nuclei subpopulations in the cumulative freezing spectrum $N_m(T)$ of lichen. The interpretability of the results in terms of subpopulations provides an advantage over polynomial fitting and differentiation of $N_m(T)$.

### 2.6 Stability of ice nucleators

Freeze-thaw cycles and heat treatments were performed on selected lichen extracts and Snomax, to assess the stability of INs exposed to temperature changes. For freeze-thaw cycles, aliquots of lichen extract were consecutively frozen by gradually cooling to -30°C at 1°C min$^{-1}$ and thawed

to room temperature twelve times over the course of twelve hours. After each temperature cycle,





the activity was determined using TINA. For heat treatment experiments, the aliquots of the lichen extracts were incubated at 98°C for one hour, and the ice nucleation activity before and after heating was determined using TINA.



**3     Results**

**3.1 IN-active lichen species**

Table 1 shows the freezing temperatures of 29 lichen extracts as determined in initial studies by a
Vali-type droplet freezing assay and measured by TINA. Freezing temperatures are shown as $T_{50}$
values and are defined as the temperatures at which 50% of the undiluted lichen extract droplets
are frozen. We find that all Alaskan lichen species collected between September 2021 and
February 2022 froze at temperatures ranging from -5°C to -14.5°C. These finding are consistent
with previous reports of widespread ice nucleation activity in lichen (Kieft and Lindow, 1988;
Henderson-Begg et al., 2009; Moffett et al., 2015; Creamean et al., 2021). The ice nucleation
activity of the collected lichen does not seem to increase from warmer (e.g., Juneau) to colder (e.g.,
Utqiagvik) climate zones and none of the lichen species collected in the Arctic tundra (Utqiagvik)
showed freezing temperatures warmer than -7.6°C. Moreover, lichens with the highest ice
nucleation activity (e.g., *Platismatia herrei)* coexist in the same coastal rainforest environment and
even on the same tree substratum as lichens with relatively low ice nucleation activity (e.g., *Usnea
longissima*). Among the 29 lichen, *Platismatia herrei*, *Peltigera britannica,* and *Sphaerophorus
globosus* showed notably high IN activity in initial screenings with $T_{50}$ values of -5.2°C, -5.7°C
and -6°C, respectively. To better characterize the INs of these lichen species, we investigated their
INs using the TINA setup, which allows the simultaneous measurement of a complete dilution
series with robust statistics (Kunert et al., 2018).






**Table 1.** Ice nucleation activity of undiluted lichen extracts determined using a Vali-type ("initial") and high-throughput ice nucleation assay ("TINA"). Freezing temperatures are shown as $T_{50}$ values and are defined as the temperatures at which 50% of the undiluted lichen extract droplets are frozen. Lichen samples were collected across Alaska, with nineteen species from coastal rainforests in Juneau (J) and three from Kodiak (K), five from the Arctic tundra in Utqiagvik (U), and two from boreal forests in Fairbanks (F). Extracts labeled N/A were not measured using TINA. Species are listed from high to low ice-nucleation activity.

| Lichen Species | Initial $T_{50}$ (°C) | TINA $T_{50}$ (°C) |
|---|---|---|
| *Platismatia herrei* (J) | -5.2 | -5.0 |
| *Peltigera britannica* (J) | -5.7 | -5.1 |
| *Sphaerophorus globosus* (J) | -6.0 | -5.6 |
| *Bryoria fuscescens* (J) | -6.2 | -5.8 |
| *Peltigera neopolydactyla* (J) | -6.3 | N/A |
| *Cladonia squamosa* (J) | -6.5 | -5.7 |
| *Hypogymnia enteromorpha* (J) | -6.6 | -5.3 |
| *Alectoria sarmentosa* (J) | -6.7 | -5.9 |
| *Cladonia cristatella* (J) | -6.7 | -6.0 |
| *Lobaria pulmonaria* (J) | -6.7 | -5.4 |
| *Platismatia norvegica* (J) | -6.8 | N/A |
| *Cladina portentosa* (J) | -6.9 | -6.6 |
| *Parmelia sulcata* (J) | -7.1 | N/A |
| *Evernia prunastri* (K) | -7.4 | N/A |
| *Stereocaulon alpinum* (U) | -7.6 | N/A |
| *Sticta fulignosa* (J) | -7.9 | N/A |
| *Platismatia norvegica* (K) | -8.0 | N/A |
| *Lobaria oregana* (J) | -8.1 | N/A |
| *Usnea wirthii* (F) | -8.2 | N/A |
| *Stereocaulon* (K) | -8.5 | N/A |
| *Dactylina arctica* (U) | -8.5 | -7.2 |
| *Cladonia macilenta* (J) | -9.2 | -7.4 |
| *Flavocetraria nivalis* (U) | -9.2 | -7.7 |
| *Xanthoparmelia cumberlandia* (J) | -9.4 | -8.8 |
| *Thamnolia tundrae* (U) | -10.3 | -9.4 |
| *Bryocaulon divergens* (U) | -11.7 | N/A |
| *Porpidia* (J) | -12.2 | N/A |
| *Usnea longissima* (J) | -12.6 | -9.8 |
| *Bryoria* (F) | -14.5 | N/A |






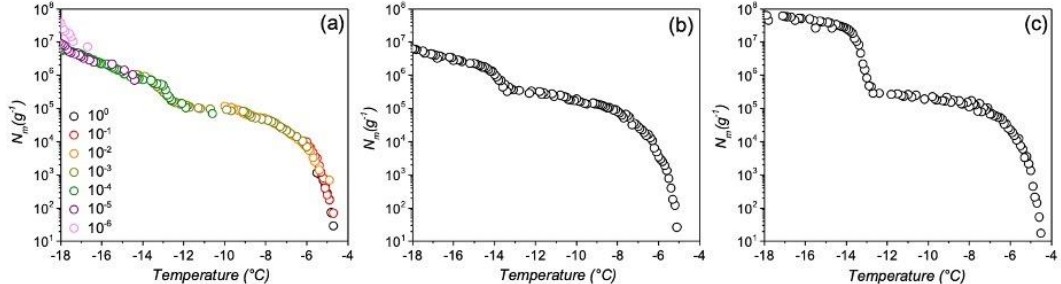

**Figure 1.** Freezing experiments of aqueous extracts containing lichen INs from *P. britannica* (a), *S. globosus* (b), and *P. herrei* (c). Symbol colors in (a) indicate data from different concentrations and are identical to uncolored dilutions shown in (b) and (c). Shown are the cumulative number of IN per unit mass ($N_m$) for extracts containing INs from lichen.


Figure 1 shows the results of TINA measurements of untreated aqueous lichen extracts of *P. britannica*, *S. globosus,* and *P. herrei*. The initial solutions had a concentration of 0.4 g/ mL and were then serially diluted 10-fold. The cumulative IN number concentration ($N_m$) was

calculated using Vali's formula and represents the number of INs per unit weight that are active above a certain temperature (Vali, 1971). We find that for the three lichen species, the freezing spectra show two strong increases in $N_m(T)$ around −4.5 and −13 °C with plateaus between ∼−7 and ∼−13 °C and below ∼−16 °C. The two rises in the spectra suggest the presence of two classes of INs with different activation temperatures, while the plateaus indicate that fewer INs are active

within those temperature ranges (Budke and Koop, 2015; Lukas et al., 2020). The presence of two IN subpopulations is notable, given that quantitative measurements of fungi-associated INs typically display only one IN population (Fröhlich-Nowoisky et al., 2015; Kunert et al., 2019). Two rises and subsequent plateaus in the freezing spectra are more characteristic of those observed in the spectrum of *P. syringae* (Fig. 3b). For *P. syringae,* the increases at ∼-2.8 and -7.5°C were

assigned to two classes of INs that consist of different aggregate sizes of the same INPs (Turner et al., 1990; Qiu et al., 2019; Schwidetzky et al., 2021a). The classes and molecular composition of lichen INs have not been identified, but it has been suggested that the fungal partner in a lichen is responsible for freezing above -5°C, while the photobiont has much lower activity (Kieft and Ahmadjian, 1989). We therefore tentatively assign the two increases in the freezing spectra of the

lichens to INs of the mycobiont and photobiont partners.





### 3.2 Underlying distribution of lichen IN

We employed the Heterogeneous Underlying-Based (HUB) stochastic optimization analysis (de Almeida Ribeiro et al., 2022) to better identify and characterize the underlying number of IN

subpopulations in lichen. Fig. 2a shows the experimentaly obtained cumulative spectrum $N_m(T)$ of *P. herrei*, together with the spectrum predicted by HUB from the optimized distribution of nucleation temperatures (e.g. the differential spectrum), shown in Fig. 2. The two IN subpopulations in the differential spectrum are centered at -6.8°C and -13.8°C (Fig. 2b). We assign the mode at -6.8°C to the mycobiont INs and the mode at -13.8°C to photobiont INs. The

differential spectrum shows that freezing at -6.8°C is two orders of magnitude less likely to happen than freezing at -13.8°C. The robustness of these signals and their physical nature require further investigation.

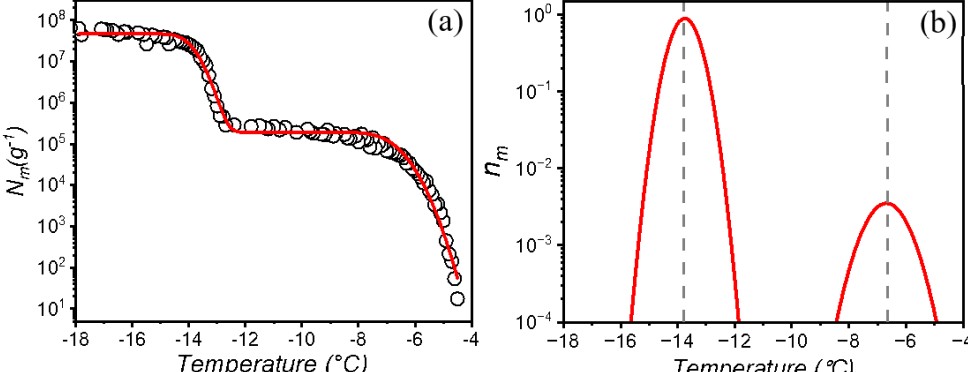

**Figure 2.** Freezing experiments of aqueous extracts containing lichen INs from *P. herrei*. (a)
Cumulative number of INs per unit mass of *P. herrei* ($N_m$) for extracts containing INs from lichen.
The red line represents the optimized solution obtained through the HUB-backward code. (b)
Normalized distribution function that represents the underlying distribution of heterogenous
freezing temperatures which gives the cumulative number of INs per unit mass. The grey dashed
lines indicate the temperatures that give the modes of the distribution. The mode, spread and
weight of the mycobiont IN subpopulation is -6.8°C, 0.67°C and 0.95%, respectively. For the
photobiont IN subpopulation the mode, spread and weight are -13.8°C, 0.29°C, and 99.05%.

### 3.3 Stability of lichen IN




In the natural environment or as aerosolized particles, the lichen INs are exposed to physical stressors such as low, high, or quickly changing temperatures. To investigate the stability of lichen INs, we performed freeze–thaw cycles and heat treatments. Figure 3 displays ice nucleation measurements of aqueous extracts of *P. herrei* and a solution of Snomax (inactivated *P. syringae*)

in water during twelve consecutive freeze-thaw cycles. We find that the freezing spectra of *P. herrei* look similar across all cycles, with nucleation events occurring consistently at ∼−4.5 and −13°C, and plateaus between ∼−7 and −13°C. In contrast to lichen INs, the freezing spectra of *P. syringae* show a systematic trend in which the final cycle differs markedly from the first. We find that the rise at ∼−2.8°C in the bacterial $N_m$ spectrum is reduced and shifted to slightly lower

temperatures with progressing cycles. The second rise at ∼-7.5°C shifts not only to lower temperatures, but also increases over time. Evidently, repetitive freeze-thaw cycles greatly reduce the activity of the class A INs (larger INP aggregates active at ∼-2.8°C) and transform them into the less active class C INs (smaller aggregates active at ∼-7.5°C). These findings are consistent with the apparent instability of bacterial INs (Polen et al., 2016), and contrast the high stability of

the lichen INs.

The repetitive freezing measurements further reveal that the highest activity of lichen IN was already developed during the initial preparation of the extracts and that no equilibration time was required. The storage of aqueous extracts at -18°C prior to freeze-thaw cycles also did not impact the ice nucleation activity of the highly active mycobiont INs given that $T_{50}$ values of the initial

measurements and up to nine months later are similar. These findings indicate that lichen INs can undergo multiple freeze-thaw cycles without losing significant activity and would hence be able to influence cloud glaciation or are reusable in freeze-tolerant lichens.

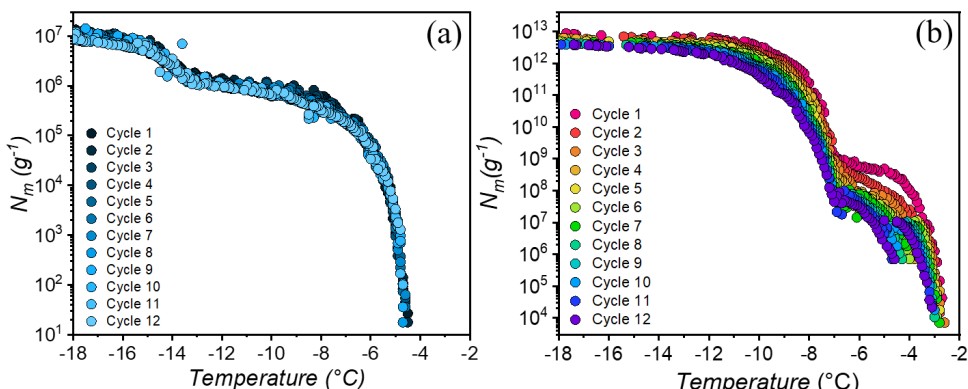

**Figure 3.** Effects of freeze-thaw cycles on bacterial and lichen IN activity. Shown are the cumulative numbers of INs per unit mass $N_m$ of *P. herrei* (a) and Snomax *(P. syrinage)* (b).

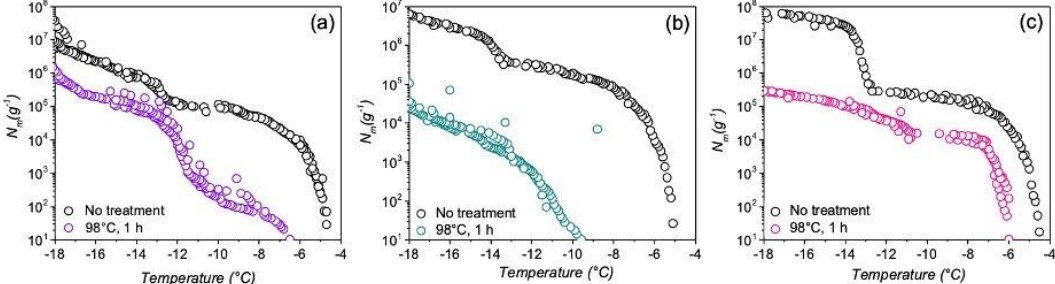

**Figure 4.** Effects of high-temperature treatment on the ice nucleation activity and corresponding IN population distribution of selected lichens. Shown are the cumulative number of INs ($N_m$) per gram of lichen sample of (a) *P. britannica*, (b) *S. globosus*, and (c) *P. herrei*.

As a second test of stability, Fig. 4 shows ice nucleation measurements of aqueous lichen extracts before and after one hour of exposure to 98°C. We find that the freezing spectra of all three lichen species are altered substantially after heat treatment. For *P. britannica* we observe that the first rise at ~-4.5 is lowered to -6.5°C, while the second rise at -13°C remains and becomes more apparent. Moreover, the total cumulative number of *P. britannica* INs decreases moderately from $10^7$ g$^{-1}$ to ~$10^6$ g$^{-1}$. For *S. globosus*, we observe a markedly different behavior: the first rise at ~ -4.5 decreases to -9.5°C, while the second rise at -13°C completely disappears. In addition, the total cumulative number of INs decreases significantly from $10^7$ g$^{-1}$ to less than $10^5$ g$^{-1}$. For





*P. herrei* we find that the first rise at ~-4.5 is only slightly lowered to -6°C while the second rise at -13°C is absent. Clearly, the lichen INs are affected differently by heat treatment, suggesting distinct variation across the biomolecules responsible for ice nucleation in lichens. We conclude that the macromolecular composition of INs of mycobionts and photobionts likely differ both within individual lichen and among lichen species. The high stability of the photobiont INs after

the 98°C treatment points toward some non-proteinaceous ice nucleating molecules, and it has been proposed that some biological INs may consist of polysaccharides (Pummer et al., 2012; Dreischmeier et al., 2017). Given the lower stability of the mycobiont INs upon heating, it is reasonable to assume that the fungal ice nucleation activity is primarily due to proteinaceous compounds that denature under high temperatures. However, future experiments, including

chemical analyses, are required to adequately characterize the macromolecules responsible for ice nucleation activity in lichen.

### 4    Conclusion

The abundance and distribution of ice nucleation activity within lichens were investigated in a study of 29 different species collected across Alaska. Among the tested lichen species, all showed IN-activity above -15 °C and ~30% showed ice nucleation activity above -6 °C, illustrating the wide distribution of IN activity within lichen in Alaska. Concentration series of the most active lichen INs in combination with statistical analysis revealed the presence of two clearly

distinguishable classes of INs which we assigned separately to the fungal and algal components of the lichen. Freeze-thaw cycles and heat treatments revealed a high stability of some lichen INs, demonstrating the suitability of lichen for potential usage in cryo-storage applications. The abundance of lichens in nature and the wide distribution of ice nucleation activity across species, together with the extraordinary stability of the INs under atmospherically relevant conditions,

indicates that these IN may have considerable impact on precipitation patterns in northern environments. We speculate that lichenized fungi and their efficient INs are likely abundant members of atmospheric communities ((Fröhlich-Nowoisky et al., 2009; Womack et al., 2015), especially over coastal rainforest canopies (Fröhlich-Nowoisky et al., 2012).

   Our results further demonstrate that there are significant differences among lichen INs and that

their molecular composition leads to variations in stability. Additional research is needed to





characterize the lichen IN to evaluate the nature of the macromolecules responsible for ice nucleation and the abundance of lichen INs in environmental samples on a global scale.





**Acknowledgements**

We are grateful to the MaxWater initiative from the Max Planck Society. K. M. acknowledges support by the National Science Foundation under Grant No. (NSF 2116528) and from the Institutional Development Awards (IDeA) from the National Institute of General Medical Sciences of the National Institutes of Health under Grants #P20GM103408, P20GM109095. I. de A. R. and V. M. gratefully acknowledge support by AFOSR through MURI Award No. FA9550-20-1-0351.

We thank Ashley Murphy, Nadine Bothen and Ralph Schwidetzky for helping with sample collection and TINA measurements and Thomas Hill and Bruce Moffett for initial discussions.

**Competing interests**. The authors declare that they have no conflict of interest.

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
