# Peer review of "Title: Lichen species across Alaska produce highly active and stable ice nucleators"

_Biogeosciences, 2022_

## Author Comment (AC1)

Eufemio and colleagues collected 29 lichen samples across Alaska. Each lichen sample was identified as a separate species and each lichen sample was found to have ice nucleation active. The most active lichens initiate freezing at -6C. Their activity is highly resistant to freeze-thaw cycles and moderately resistant to heat treatment. Interestingly, two classes of ice nuclei were found. They are active at different temperatures. The authors assume that the nuclei active at higher temperature are from the mycobiont component and the nuclei active at lower temperatures are derived from the photobiont component.

Overall, this manuscript is very well written and interpretation of results and conclusions are overall well justified. I only have two major comments:

**Response:** We thank the reviewer for carefully reading our manuscript and address each comment, point-by-point, below.

1. The authors do not provide methods on how the 29 lichens were taxonomically identified. Details of how the identification was made need to be included in the methods section.

**Response:** We thank the reviewer for raising this important point. We sampled lichens based on availability in the natural environment and feasibility of access to collection sites. We exclusively sampled foliose and fruticose lichens since these species protrude off their substrates in leaf- and hair-like structures, which enables accurate identification based on physical features.

**Action taken:** We added additional text and a reference to the methods part of the manuscript. The text begins at page 4, line 95, and reads, "Lichen samples were collected based on availability in the natural environment and feasibility of access to collection sites. We exclusively sampled foliose and fruticose lichens. These species protrude off their substrates in leaf- and hair-like structures, which enabled accurate identification based on physical features. Species were identified using a vegetation identification guide (Pojar and MacKinnon, 1994)."

2. The authors appear to make a leap when assigning the more active nuclei to the mycobiont and the less active nuclei to the photobiont. However, to me it seems as likely that both classes of ice nuclei consist in the same molecule (be it a protein or something else) derived from the mycobiont and the classes are simply due to different aggregate sizes of the same monomeric molecule produced by the mycobiont. Since there seems to be no experimental results pointing to either the authors hypothesis or the hypothesis I propose here, I would not refer to these classes as mycobiont and photobiont in the manuscript. I would only speculate that they may and then just refer to them as classes A and B or some other neutral naming in the rest of the manuscript.

**Response**: We thank the reviewer for bringing our attention to this point. We revised the manuscript to clarify that the experimental results of Kieft and Ahmadjian (1989) have previously identified the mycobiont as more ice-nucleation active than the photobiont. We therefore speculated that the two increases in the freezing spectra of the lichens can be attributed separately to the mycobiont and photobiont partners. However, it is also possible that the lichen INs are exclusively mycobiont-derived macromolecules that aggregate similar to bacterial ice-nucleating proteins and induce freezing at different activation temperatures. Given the uncertainty, we

followed the advice of the referee and referred to the classes as class 1 and class 2 in the rest of the manuscript.

**Action taken**: We amended the sentence starting on page 9, line 223, to read, "The classes and molecular composition of lichen INs have not been identified, but experimental results of Kieft and Ahmadjian (1989) have previously identified the lichen mycobiont as responsible for freezing above -5°C, while the photobiont has much lower activity. We therefore tentatively assign the two increases in the freezing spectra of the lichens to INs of the mycobiont and photobiont partners. However, we cannot exclude that lichen INs are exclusively mycobiont-derived macromolecules that aggregate similar to bacterial INPs, and thus induce freezing at different activation temperatures depending on their aggregate size. Given the uncertainty, we refer to the IN subpopulation active at $-4.5°C$ as class 1 and the second subpopulation active at -13°C as class 2."

Minor comments

1. The sentence in lines 81-83 could be improved. "to gain insight into possible atmospheric influences" is vague and it could either mean influence of the lichens on the atmosphere or influence of the atmosphere on the lichen". Please rephrase.

**Response:** We thank the reviewer for the suggested improvement. The text has been revised to clarify that we investigated the stability of lichen INs to gain insights into possible atmospheric implications of airborne lichen INs.

**Action taken:** We rephrased lines 81-83 of the manuscript. The text now reads, "…to gain insights into possible atmospheric influences of airborne lichen INs…".

2. Line 58, I would specify how many of the 29 lichen samples were analyzed using TINA in this sentence.

**Response:** We thank the reviewer for the comment and revised the text to specify that 16 of the 29 lichen samples were also analyzed using TINA.

**Action taken:** The text was revised and now reads, "Table 1 shows the freezing temperatures of 29 lichen extracts as determined in initial studies by a Vali-type droplet freezing assay and 16 of the lichen extracts as measured by TINA."

3. I am just wondering how the sampling was done. Since each sample turned out to be a different species, I guess that the authors specifically looked to find a different species at each sampling site? If sampling was random, I would have expected that the same lichen species would have been found more than once. I suggest to clarify the sampling strategy to make it easier to understand why each sample turned out to be a different species.

**Response:** We thank the reviewer for addressing this point. We exclusively collected fruticose and foliose lichens. Among these, we did not purposefully collect specific species but rather selected those that were easily identifiable based on their external morphology.

**Action taken:** We clarified this point on page 4, lines 96-99 of the methods section. The text reads, "We exclusively sampled foliose and fruticose lichens. These species protrude off their substrates in leaf- and hair-like structures, which enabled accurate identification based on physical features".

4. I wonder if the authors could comment on what kind of ice nuclei the grinding of the lichens released. I guess that the method was chosen to include both cell wall-bound non-secreted molecules as well as secreted molecules. However, I think it would be good to add a sentence somewhere specifying what the authors expected to be in the tested samples: only secreted molecules or all molecules independently of being secreted or not.

**Response:** We revised the methods section of the manuscript to clarify that the procedure was chosen to ensure that the aqueous extract contains all molecules that are present in the lichen samples (e.g. macromolecules bound to the cell wall and secreted molecules).

**Action Taken:** We added text on page 4, line 109, which reads, "This procedure was chosen to ensure that the aqueous extract contained all the molecules present in the lichen samples, both the molecules bound to the cell wall and any secreted molecules."

---

## Author Comment (AC2)

The present paper on ice nucleation activity is a thoroughly done study of the ice nucleating actity of 29 lichens found across Alaska. Homogenates of all of these show ice nucleating activity above -15 degC. Some even at relatively high temperatures at around -5 degC to -6 degC. the authors have made some preliminary experiments trying to narrow down the nature of this IN activity and compellingly show that in some of the species the activity is presumably due to proteinaceous ice nucleators whilst in others the IN activity is possibly due to polysaccharides or other non-proteinaceous substances as their IN activity not changing much after heat treatment. In general, it is surprising how stable the ice nucleation substances are which is in stark contrast to the rather instable bacterial INAs.

It is interesting that it does not seem as if the INAs in the lichens are related to the severity of the low temperatures in their habitat but possibly to other parameters - one could speculate about the humidity or water logging of the habitat. Another possibility could maybe that the occurrence of INAs in lichens is just an intrinsic property and not of any adaptive value in relation to low temperatures. This problem is not discussed by the authors. It could thus be interesting to test lichens from warmer climates to see if these also are showing IN activity.

**Response:** We thank the reviewer for carefully reading our manuscript, the positive comments, and the suggestion to test lichens from warmer climates. As pointed out by the reviewer, the important question about the ecological benefits of ice nucleation activity for lichen from an ecological perspective remains unknown. However, whether ice nucleation activity in lichen is an intrinsic property, provides advantages for water logging, or for freeze-tolerance cannot be answered with the available evidence and is outside the scope of the current work. However, we plan to systematically address the connection between ice nucleating activity and geographical distribution in lichen in future research.